# The Fragility of Women's Work Trajectories in Chile

**Rosario Undurraga * and Jóna Gunnarsson**

Faculty of Education, Psychology and Family, School of Family Sciences, Universidad Finis Terrae, Santiago 7501015, Chile; jona.gunnarsson.fo@gmail.com
* Correspondence: mrundurraga@uft.cl

**Abstract:** How are the work trajectories of Chilean women? This qualitative study analyzes the female work trajectories through interviews and biograms in a sample of 50 Chilean women, professionals and non-professionals, between the ages of 24 and 88. The article proposes an original typology of female work trajectories and relates type of work trajectory with Piore's theory of labor market segmentation. The paper discusses the challenges and weaknesses of the Chilean women's labor outcome and presents recent data to extrapolate the impact of the COVID-19 pandemic on vulnerable work trajectories. It considers the type of State and possible actions to achieve greater welfare and social development regarding gender equality.

**Keywords:** work trajectory; women; COVID-19; gender inequalities; care

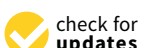



## 1. The Chilean Labor Market

Work and family play a central role in Latin America and in Chile (Inglehart et al. 2014, WVS-6). Employment opportunities for women and their participation in labor force have increased during the last decades, however, the Chilean female participation rate in the labor market is still low (46.1%) compared to men (69.1%) and women elsewhere (INE 2021a, 2021b). Cultural factors play an important role in understanding the low participation of women in the Chilean labor market (R. Undurraga 2013; Contreras and Plaza 2010). Gendered expectations and the traditional role of women are projected onto both the private and public spheres, and social constructions of womanhood impact on women's daily lives, in the debate over public policies and regulations. The highly unequal context hinders women from participating in the public arena as women are primarily expected to perform caring roles (R. Undurraga and Hornickel 2021; Mora and Blanco 2018). As such, women experience severe discrimination in the labor market including maternity penalties (R. Undurraga 2018, 2019a; R. Undurraga and Barozet 2015). There are relevant differences between men and women's salaries and career opportunities, represented in high occupational segregation, and low quality jobs for women (INE 2018; Cárdenas 2018). Women are overrepresented in the informal sector and within it they are more likely to be found in precarious and less well paid jobs. Women in the informal sector are largely excluded from formal social protection (INE 2020a; Barrientos 2004).

The social security and pension system in Chile has its origins in Pinochet's military dictatorship (1973–1989), when a series of reforms related to the neoliberal model were implemented (Araujo 2017; T. Undurraga 2014; Gárate 2012), including labor flexibility, restrictions on collective bargaining, and trade union's repression (Acuña 2008). Neoliberalism led the country to a liberal residual welfare state, privatizing social security and reducing the role of the State to a more subsidiary one (Larrañaga 2010). For example, the pension system is based on individual capitalization: a mandatory personal savings of 10% of people's salary administered by the Pension Fund Administrators (AFP). Women receive significantly lower pensions than men due to the design of the system and factors unique to the individual—gender, lower salary, fewer number of years worked, earlier retirement age (R. Undurraga and Becker 2019; Comisión Asesora Presidencial Sobre el Sistema de

Pensiones 2015), greater work informality (INE 2020a), lower levels of contributions, and greater life expectancy (Superintendencia de Pensiones 2020). The amount of the pension depends on the risks associated with the population and economic variations in the market. The pension and health systems reproduce the gender gaps observed in the private and public sphere related to the traditional division of labor.

How are the work trajectories of Chilean women like? This article shows an original typology of female labor trajectories and uncovers the fragility of most women's work path in Chile. It extrapolates the results to women's labor outcome during the COVID-19 pandemic. The research takes an intersectional approach to understanding multiple dimensions of inequality in the Chilean labor market (Mora 2019; R. Undurraga 2019a). Intersectionality assumes that social categories relate to each other, overlap, and intertwine; they are not separate (Ferguson 2013; McCall 2005; Crenshaw 1991). This means that by using an intersectional analysis, gender, age, education, marital status, among other social categories, interact and relate to each other within and across different contexts revealing privileges and vulnerabilities.

## 2. Methodology

This qualitative study explored the work trajectories of Chilean women. An original typology of women's work trajectories is presented here. The theory of labor market segmentation by Piore (1972) is used as a theoretical explanation for these trajectories. In addition, in the context of the COVID-19 pandemic, the link between the type of work trajectory and the possible consequences regarding female work trajectories is discussed.

### 2.1. Sample

The sample contained 50 women residing in Santiago de Chile—25 professionals and 25 non-professionals—with and without paid work, between the ages of 24 and 88, equally distributed in five age groups: 20–35, 36–45, 46–60, 61–75, and 76 and over. Each group was made up of five professionals and five non-professionals. Professionals refer to women with a terminal university degree, which in Chile consists of approximately five years of study, who are engaged in professional tasks in the labor market. Non-professionals do not have a university degree and carry out jobs that do not require higher education.

An intentional sampling was carried out following convenience criteria and bearing in mind the inclusion criteria described above for each group: sex, age, and region of residence. Key informants, contacts from the research team, and the snowball technique were used to access the participants, who were contacted by the researchers to arrange an interview.

Regarding the occupational situation, some participants were active in the labor market (with and without paid work) and others were retired, but all of them have had at least one paid job in their working life. At the time of the interviews, 16 of the 25 professionals and 19 of the 25 non-professionals had paid work (70% working for a wage and 30% outside the labor market). The professionals worked as teachers, engineers, sociologists, dentists, veterinarians, medical doctors, designers, and psychologists, among other professions. The non-professionals worked as receptionists, secretaries, cashiers, accounting analysts, pollsters, saleswomen, and domestic workers, among other jobs. Of the sample, 25 worked for a wage, 10 worked independently, and 15 did not have paid jobs (see Table 1).

Concerning their personal and family situations, among the 50 women interviewed, 36 were mothers. Regarding marital status: 14 were married, 5 cohabited, 13 were single, 9 were widows, 6 were separated, and 3 were divorced. Regarding the number of children, on average they had 2 children, between 0 and 10 children. Of these, 13 participants had children under 18 years of age (see Table 2).

**Table 1.** Occupational situation according to age, by professionals and non-professionals.

| Age Group | Professionals | | | Non-Professionals | | | Total |
|---|---|---|---|---|---|---|---|
| | Waged Worker | Independent Worker | Out of the Labor Market | Waged Worker | Independent Worker | Out of the Labor Market | |
| 20–35 | 3 | 1 | 1 | 4 | 0 | 1 | 10 |
| 36–45 | 4 | 0 | 1 | 2 | 3 | 0 | 10 |
| 46–60 | 4 | 1 | 0 | 3 | 2 | 0 | 10 |
| 61–75 | 2 | 1 | 2 | 3 | 1 | 1 | 10 |
| 76+ | 0 | 0 | 5 | 1 | 0 | 4 | 10 |
| Total | 13 | 3 | 9 | 13 | 6 | 6 | 50 |

**Table 2.** Number of children according to educational attainment.

| Number of Children | Professionals | Non-Professionals | Total |
|---|---|---|---|
| 0 | 9 | 5 | 14 |
| 1 | 0 | 7 | 7 |
| 2 | 6 | 6 | 12 |
| 3 | 6 | 3 | 9 |
| 4 | 0 | 4 | 4 |
| 5 | 2 | 0 | 2 |
| 6+ | 2 | 0 | 2 |
| Total | 25 | 25 | 50 |

*2.2. Materials and Procedure*

In order to collect information on the women's work trajectories, a semi-structured interview was used together with a biogram, which is a graphic representation that each participant makes of her work history. The interview delved into the milestones highlighted in the biogram. The conversation was structured according to the moments that the participants defined as relevant throughout their working lives, investigating the working conditions of each job, reasons for entering and leaving, experiences, and relationships between work and family life, among other aspects.

The interviews were conducted between April 2016 and May 2017 in the most convenient place for each participant (their home, a coffee shop or their workplace) and lasted between 35 min and 2 h 30 min (average 1 h 21 min).

All interviews were carried out voluntarily, and after signing informed consent that ensured the protection of the participants' identity. They were informed about the objectives of the study, guaranteeing them the conditions of voluntariness, anonymity, and confidentiality in the handling of the data. When using excerpts from the interviews, the participants are identified with a number (ID), indicating age, profession/occupation, relationship status and number of children. This research has the approval of the Ethics Committee of the Universidad Finis Terrae and CONICYT/FONDECYT.

*2.3. Analysis Strategy*

The interviews were recorded and transcribed (verbatim). For the analysis and coding process, Atlas.ti software was used as a support tool to raise codes and categories and graph relationships (Berlin, Germany, 1993). The information was analyzed based on grounded theory (Glaser and Holton 2004; Glaser and Strauss 1998). This implies that in the analysis process, emerging categories were noted (Krause 1995), and a comparative exercise was performed that allowed the coded data to be connected. The 50 interviews were analyzed creating categories and subcategories, to finally create a conceptual network through maps that allowed organizing the information, selecting the topics related to the research questions, and gathering the main findings.

This methodology implies an iterative process, which allows space to reach a level of theoretical saturation, an instance in which new data does not add new information (Glaser



and Holton 2004; Krause 1995). Using this methodology, we proposed an original typology of Chilean women's work trajectories and discussed the challenges and weaknesses of the labor outcomes of these women.

## 3. Results and Discussion

### 3.1. Typology of Female Work Trajectories

"The analysis of labor market is an important concern for sociology inquiry; it permits an understanding of the way macro forces associated with the economy of a society and elements of social structure impinge on the microrelations between employers and workers in determining various forms of inequality" (Kalleberg and Sørensen 1979, p. 1). As an explanatory approach to the discrimination of minorities in labor markets, Piore (1972) elaborated the theory of labor market segmentation. He focused on male black workers' socio-economic mobility (or lack of) in the United States. He presented a dual labor market compound by two segments: the primary and the secondary. The primary sector is characterized by high wages, good working conditions, chances of advancement, and job stability. In contrast, the secondary sector is characterized by low paying, poor working conditions, and little chance of advancement. This theoretical framework helps the understanding of the categorization of Chilean women in the labor market, as the working context resembles social inequalities and discrimination to minority groups. Women in Chile experience gender, class, marital status, and age discrimination in the labor market, which hit their work trajectories (R. Undurraga and Hornickel 2020, 2021; R. Undurraga 2019a). More than half of women in Chile do not have paid work (INE 2021b).

We proposed an original typology of Chilean women's work trajectories, identifying four types: (1) self-realization work trajectory, (2) assured-work trajectory, (3) work trajectory depending on the family, and (4) improvised-work trajectory.

These trajectories are connected to sociodemographic factors, such as educational level, socioeconomic level, age, marital status, and children, embracing an intersectional approach to unfold inequalities in the labor market. Belonging to one type of trajectory is not necessarily permanent throughout a woman's life. The type of trajectory can change and, therefore, the work trajectories are not static or predictable; they are dynamic. It is possible to change from one type of trajectory to another as a result of variations in the mentioned sociodemographic factors or due to external events, such as a crisis like the COVID-19 pandemic. In this sense, we highlight the relational nature of labor trajectories, where women's relationships and the context play an important role in their definition and changes. At the same time, it is not possible to speak of women's work trajectories in general; this typology shows the diversity and heterogeneity in the labor trajectories of Chilean women.

Below we describe the four types of work trajectory and relate them to Piore (1972) theory of labor market stratification.

(1) The self-realization work trajectory is mainly represented by young professional women (20 to 35 years old), who are single or cohabiting, without children. This trajectory is characterized by focusing on the personal goal of self-realization. Work is a means to achieve this. Thus, it predominates a utilitarian vision of work, where current work is conceived as a springboard to progress toward a personal goal, with an average length of two years in each workplace. This is expressed by a 29-year-old history graduate, single, without children (ID4) who said, "work was not about achieving great things at work, for me it was a means to achieve certain goals for my personal life . . . I didn't feel that work was going to give me identity today."

The valuation of work is based on the professional and/or personal development rather than on working conditions. Although it is considered ideal to have a long-term contract, which provides job stability and social security, it is common in this type of trajectory to be paid by the hour or have a short-term contract. The completion of a post-graduate degree is considered important as a part of their personal goal to continue growing and improving throughout their professional careers.

Part of the work strategy pursued in this type of trajectory is to improve their curriculum in order to enhance the chances of getting desirable jobs in the future, without having a specific goal in the present. Networking is considered crucial to stand stronger in the labor market, possibly as a strategy against the job instability that often is assumed as a part of this type of trajectory. In this sense, a strong "presentism" predominates among young women (R. Undurraga and Becker 2019; Leccardi 2014), where having a clear and specific plan for the future may seem obsolete; therefore, it is better to imagine an open future and focus on the present. This is how a 28-year-old biology teacher, cohabiting, without children narrates it:

> I find that you have to be really comfortable with what you do, and not just do it thinking about a sacrifice . . . you don't live to work . . . I don't define my life based on work either, I mean, for me it is super important, but tomorrow it may occur to me to stop doing this and do something totally different. (ID1)

The self-realization trajectory resembles Piore's upper tier of the primary labor market, containing more high profile jobs such as manager positions and other professional work positions. The jobs in the upper tier of the primary labor market are usually well paid and imply some kind of status and frequent promotion opportunities. Regular turnovers are normal and a part of advancing in the right direction, showing great mobility among the workers in the upper tier and low risk for descending in their careers when experiencing layoffs. These jobs contain less rules and formalities and allow more individual creativity and initiative making space for self-realization (Piore 1972).

(2) The assured-work trajectory is made up of professional and non-professional women, mainly over 40 years of age, mostly married with children. The main priority is that work is compatible with family life; therefore, job expectations are focused on the working conditions—the schedule, the contract, and the salary. If the working conditions are good and they have achieved job stability, that is enough for these women to project themselves in that same job until retirement; subsequently, there is little rotation (more than 10 years in each job). The main work strategy consists of recognizing and complying with the requirements of the specific institution where they work in order to get promoted or to improve their working conditions. For this purpose, there is a deeper integration of the women with their workplace, and a continuity that gives rise to certain conflicts that need to be resolved in order to continue in that work. A 42-year-old analyst in an insurance company, married, with three children commented:

> I feel that they pay me well and my boss makes sure that I am happy because I have been there for so long and I know all the goings-on in the area, so every time I want my salary to go up, I tell him that I'm going to change to a different area, and he fixes my salary so that I can stay . . . I am very comfortable where I am, happy. (ID19)

The assured-work trajectory is similar to the lower tier of the primary labor market of Piore's distinction having stability and routinized jobs at its core. Receiving higher education is not a goal in itself but a way to more easily accede well paid jobs that assure economic stability for the workers and their families—a key in entering the primary labor market. However, higher education in the lower tier is not an essential requisite as it is in the upper tier, since informal training and work experience can substitute formal education. This also has a downside in terms of layoffs, as human capital in this group is firm-specific and therefore the risk of falling into the secondary labor market is higher.

(3) The work trajectory depending on the family occurs primarily among young women (20 to 35 years old), non-professionals, married or cohabiting, with children. In this type of trajectory, work is seen as a complementary activity, which women can accept or reject as the family situation allows, since women's main responsibility is that of caregivers who must be available to her family needs. The working conditions are substandard (low salary, shift hours, working weekends), as there is much rotation between different jobs, low qualifications, and frequent exits from the labor market, all of which reduce their

chances of advancing in their work trajectories. The most valued aspect of their labor participation is economic independence. It is also agreed that working outside the home contributes to personal and emotional well-being and presents a space for partaking in social events. For these reasons, working conditions are not assigned much importance, since the mere fact of working and leaving home is seen as a benefit. A 29-year-old seller, cohabitating, with three children explained:

> I started now because, well, firstly, because I was already bored at home and secondly, because my son had finally started full-time [at school] . . . For twelve years I was without working, at home . . . spending all day at home, the only thing that you do is wash, iron, clean, make lunch . . . so it's a constant routine. (ID10)

This quote shows that her dedication to housework was not entirely satisfactory, but that now that her children were experiencing a full school day (due to their age), she was able to work without compromising her duty as mother.

Piore based his theory on male workers in the United States in the seventies and therefore the caretaking chores/responsibilities are not included in his equation and it is harder to find the direct resemblance to a work trajectory which accounts for family. However, the work characteristics found in this trajectory are those of the secondary labor market "low-paying, with poorer working conditions, little chance of advancement; a highly personalized relationship between workers and supervisors which leaves wide latitude for favoritism and is conducive to harsh and capricious work discipline; and with considerable instability in jobs and a high turnover among the labor force" (Piore 1972, p. 2). The difference is that in this trajectory, labor participation is adjusted to the families' needs of the woman.

(4) The improvised-work trajectory is dominated by non-professional women of all ages. In many cases, they are single mothers who live with their children in their parents' home. This trajectory is characterized by the absence of a work strategy that guides the decisions concerning their work trajectory, being more tactical and operating without agency and without planning, which creates more vulnerability.

Vulnerability in this context can be defined as the extent to which an individual can anticipate, cope with, resist and recover from the impact of human, physical, environmental or economic hazards that are detrimental to well-being and to afford a living. People differ in their exposure to risk as a result of their gender, age, educational attainment, and other factors, which affect their work path, salary, and job stability.

The permanence or non-permanence of a job is determined by the working conditions of that job, along with external factors such as dismissal or layoffs. Given that the working conditions that exist in this type of trajectory are not good, the duration of each job is typically very short (from a couple of months to a couple of years). This is how a 24-year-old analyst, cohabitating, with one child recounted her experience:

> Like when I can't take it anymore, I leave or they lay me off or they go bankrupt . . . it's a little unstable, because I'm 24 years old, and I've gone through five jobs plus an internship, which would be six. So, it's kind of unstable. (ID7)

The jobs that are accepted in the improvised trajectory do not provide opportunities for the future, but only solve the economic needs of the moment. For this reason, the women also fail to create networks that can lead to better quality jobs. The duration of unemployment that occurs between jobs is usually short, since they require an income every month, so they will take the first opportunity that emerges (there is no strategy for the future). Additionally, some employers have practices that exploit these women by demanding longer working hours without paying overtime, and promising certain benefits without delivering them, among other abuses. For example, a 38-year-old secretary, separated, with two children stated:

> . . . apart from that I do more than what . . . I mean, I sign a book saying that I enter at 9 in the morning and leave at 6:20 in the afternoon, Monday to Friday

and on Saturday I work from 9.00 to 12.00. But my actual hours are from 8.30 to 6:30 in the afternoon, and on Saturday I work until 1:00 p.m. Therefore, I give away for free almost half a day from the schedule. Then, on top of that staying late without being paid. (ID20)

The difficulties that arise in this improvised-work trajectory, such as physical and mental exhaustion, which are products of poor working conditions, emerge and limit the work trajectory. Exhaustion is especially noticeable in trajectories where the body is the essential work tool, as it is for domestic workers. A 68-year-old domestic worker, widow, with four children (ID38) who continues working after her retirement age to pay her bills, remembers working for several families in one week, highlighting the physical fatigue and the long working hours: "I came home in the night, but no need to tell you how my feet were at that point." On the other hand, a 29-year-old secretary, cohabiting, without children narrated the harassment she suffered at her work:

I was there for a very short time. I was there for about two months . . . because there were a lot of mine-diggers or construction workers, who were nasty, lascivious, and I couldn't take it, I collapsed. It was a lot of stress. Because you can't treat them badly or answer them harshly and like I collapsed. That was like the difficulty I had there, that's why I quit. (ID6)

Although gender violence at work crosses labor trajectories of many professional and non-professional women in Chile (R. Undurraga and Hornickel 2020), in this improvised-work trajectory, women are more exposed and vulnerable to abuse since they have fewer educational, cultural, and labor resources to address gender inequalities. At the same time, they deal with frustration and the feeling of having experienced injustice. There is no way to prove the increase in their knowledge, as they do not have formal qualifications; what they have learned, they have acquired in the course of their careers, without formal study. Finally, it is very difficult to reconcile work and family life due to these working conditions and the poor economic resources for seeking private care alternatives. Therefore, grandparents play an important role in taking care of grandchildren. For example, a 43-year-old pollster, single, with one child (ID18) points out: "I can peacefully work; I always count on my mom and dad to take care of my daughter. Excellent, without them I couldn't [work]."

This type of trajectory derives from the social and economic vulnerability of these women and from their families, for whom they require quick solutions to urgent economic needs. The complicated fact is that this type of trajectory produces vulnerability, which generates a life-long cycle in which women lack sufficient social and economic resources to escape this situation of vulnerability. These women have to reconcile motherhood (taking care of their children) with the conditions of these jobs (long and/or night shifts and weekend work), based on their family of origin, since the institutional offerings of the State for low-income families do not meet their needs, and do not offer the resources required to pay a private person to take care of the people depending on them. For these women, living in their parents' homes is often their only option. This is consistent with the findings of Palma and Scott (2020) on housing solutions for young and low-income mothers who work, wherein extended family living is increasingly important in the early stages of family formation. They show an increase in intergenerational dependence, which is mainly driven by the economic and social support required by the young adult generation.

The improvised-work trajectory is very well defined by the secondary labor market proposed by Piore (1972), which is mentioned in the trajectory depending on the family. What is very interesting, is that the labor segmentation theory was in part a way to explain the high unemployment rate among the vulnerable population (black men) as a result of instability and high turnovers, characteristics of the secondary labor market, more than actual unemployment. This is similar to the results of a study performed in Chile addressing the low female participation rate in the work force finding a greater instability in their work trajectories rather than no participation at all (PNUD 2010).

Table 3 below summarizes the classification of the participants according to types of trajectories regarding educational level (professional or non-professional), age, marital status, and number and age of children, showing the trend in each type of trajectory. The self-realization work trajectory is mainly composed by young professional women that are single and without children; the assured-work trajectory is characterized by married, middle aged women, both professional and non-professional, with around three to four children older than five years old; the work trajectory depending on the family is composed by young non-professional women, married or cohabiting, with one to two children; and the improvised-work trajectory contains non-professional women up to retirement age, that are single mothers with one or two children. This shows clearly that age, educational level, marital status, and family situation play a great role in defining the type of work trajectory that Chilean women are a part of. In order to belong to the higher tier of the primary labor market, women should be young, single, professionals without children and for the women that do not meet these requirements, they will have to settle for less success, stability, and lower salary. It is also noteworthy, that these factors do not operate independently, but instead are related in a way where certain combinations seem to be more disadvantageous than others. Being single can be a positive factor for the young women belonging to the self-realization work trajectory, since it allows them to devote time and effort to work in a culture that otherwise would expect them to dedicate more of themselves to their husbands and children. On the other hand, in the case of the women that have children, being single is a factor that negatively affects their work-life.

**Table 3.** Distribution (%) of age, educational attainment, marital status, and motherhood situation in each type of work trajectory.

| Type of Work Trajectory | | Self-Realization Work Trajectory | Assured Work Trajectory | Work Trajectory Depending on the Family | Improvised Work Trajectory | Retired | Total |
|---|---|---|---|---|---|---|---|
| Education | Profesional | 100% | 47% | 17% | 0% | 62% | 50% |
| | Non-profesional | 0% | 53% | 83% | 100% | 38% | 50% |
| Subtotal | | 100% | 100% | 100% | 100% | 100% | 100% |
| | 20–35 | 56% | 0% | 50% | 29% | 0% | 20% |
| | 36–45 | 22% | 27% | 33% | 29% | 0% | 20% |
| Age group | 46–60 | 22% | 27% | 17% | 43% | 0% | 20% |
| | 61–75 | 0% | 40% | 0% | 0% | 31% | 20% |
| | 76+ | 0% | 7% | 0% | 0% | 69% | 20% |
| Subtotal | | 100% | 100% | 100% | 100% | 100% | 100% |
| | Single | 44% | 13% | 0% | 43% | 31% | 26% |
| | Divorced | 22% | 20% | 17% | 29% | 8% | 18% |
| Marital status | Cohabiting | 11% | 0% | 33% | 29% | 0% | 10% |
| | Married | 22% | 47% | 50% | 0% | 15% | 28% |
| | Widowed | 0% | 20% | 0% | 0% | 46% | 18% |
| Subtotal | | 100% | 100% | 100% | 100% | 100% | 100% |
| Age of children | 1 or more children <6 years old | 0% | 7% | 50% | 29% | 0% | 12% |
| | Only children >5 years old | 33% | 73% | 50% | 43% | 77% | 60% |
| | No children | 67% | 20% | 0% | 29% | 23% | 28% |
| Subtotal | | 100% | 100% | 100% | 100% | 100% | 100% |
| | 0 children | 67% | 20% | 0% | 29% | 23% | 28% |
| Number of children | 1–2 children | 22% | 27% | 67% | 71% | 31% | 38% |
| | 3–4 children | 11% | 47% | 17% | 0% | 31% | 26% |
| | 5+ children | 0% | 7% | 17% | 0% | 15% | 8% |
| Subtotal | | 100% | 100% | 100% | 100% | 100% | 100% |

Source: Elaboration from the interviews carried out.

It should be noted that, in Chile, the legal retirement age for women is 60 years. In addition, among the women who should be retired, half of the non-professionals continue working and belong to the assured-work trajectory, and of the professionals, two out of ten elderly women belong to that trajectory. This shows that (a) pensions are not enough to afford a living; (b) women who continue working after the retirement age due to economic

necessity, do so in the same kind of job as before, reproducing the labor logic in retirement; and (c) the legal retirement age represents an external requirement rather than a decisive milestone to finalize their work trajectories.

> I worked at [university] for 17 years as cleaning and administrative assistant. I am 69 years old. I started selling chocolates, cheese and olives seven years ago when I retired; the Director allowed me to stay around selling these things. So I make a living based on what I get here and with the little pension, because it is not much, it is $152.000 CLP. (ID40)

The experience of our participants is articulated with national data, which shows that about one third of elderly people continue working and the main reason to do so is economic necessity (Centro UC Estudios de Vejez y Envejecimiento 2017). The average amount of pension for women is $214.644 CLP, which is equivalent to US$ 305 (US$ 1 = $702 CLP [18 April 2021]), while $353.133 CLP for men, with a gender gap of 39.2%; the median amount of pension is $148.647 CLP for women and $241.337 for men (Superintendencia de Pensiones 2020). Women's pension amount is lower than the minimum wage, which currently is $326.500 CLP, and the median is below the poverty line ($176.625 CLP per person) (Ministerio de Desarrollo Social 2021).

### 3.2. Dynamism and Vulnerability within Labor Trajectories

We have presented four types of work trajectories for Chilean women. Despite the fact that the typology presented shows diversity and heterogeneity, several of them face severe challenges because of the vulnerability that emerges in times of crisis. Trajectories can be dynamic, depending on changes due to factors in the private sphere, such as a change in the family configuration (having children, marring, divorcing, etc.), and the transitions from one type of work trajectory to another can also be due to external factors that, without prior notice, affect the context such that the work trajectory changes without offering women another alternative.

The quote below from participant ID18, a 43-year-old, pollster, single, with one child, shows the dynamism of Chilean women's work trajectories intertwined with traditional gender roles, tainting her trajectory with vulnerability. The loss of a job in some cases means changing from one kind of work trajectory to one of greater fragility.

> My thing, that was going well, very well, but in order to raise her [child] . . . I was an agricultural technician, they sent me to different places in Chile and my thing was going very well. But when she was born, I couldn't do that anymore, because I had a little girl, a baby, so I gave up working there. And I went to work in the sales world, which I did not like at all because it is a world that is a lie, a lie. You have to lie to attract customers, and I didn't like that very much. It still met my income expectations, because in sales you earn good money, but then there was an economic downturn, and there I had to talk to the employer to get them to lay me off. And not being able to do anything and with a super high Dicom [debt] I had to give up work to pay a little bit of those debts [with the settlement], and I was still in Dicom, I had no options to look for a good job and I threw myself to sell on the street, lunches, without that I have nothing else to do. It was my only option at that moment . . . And that was a job that marked my life a lot, because it was very sad because I was faced with something that I never thought I was going to do, which was working on the street selling lunches . . . Humiliations . . . The police approached you with great arrogance, they took you to jail, they threw away your merchandise, and you still had to bring home the food, since I had a little daughter. (ID18)

This narration shows the decay of a self-realized work trajectory, which after four years, due to the birth and raising of her daughter, changed to an assured-work trajectory. Ten years later, a great scam occurred at her workplace; as a result of that and her debts, she ended up in an improvised-work trajectory, with little possibility of ascending again.

She summarized her work history like this and we add the respective type of trajectory: "I started from super high [1: self-realization work trajectory], like super stable [2: assured-work trajectory] and I am at this minute super unstable [4: improvised-work trajectory]." This quote illustrates the fragility and vulnerability that exist within the labor trajectories of women in Chile, since there is no social security that protects them against external events.

Notwithstanding the decline in the work paths, some participants, like domestic workers currently from the assured-work trajectory, started on the improvised-work trajectory, and then experienced improvement in their working conditions, well-being, and job stability. Also, a few professionals started and ended up on the self-realization trajectory. However, most participants moved within work trajectories according to their families' needs.

Piore (1972) states that there is a much higher mobility within the primary labor market, both in the upper and the lower tier, progressing towards higher salaries and status, while in the secondary labor market there is no real progression since the workers are easily substituted by other workers and the human capital is not enhanced by their jobs. This explains partially why it seems so hard to progress from a trajectory belonging to the secondary labor market to one belonging to the primary labor market.

The work trajectories of Chilean women are diverse (some more dignified than others), but in the context of poor social protection, they all hang by a thread that can be easily cut in the face of a crisis like the pandemic that we are experiencing. This exemplifies the fragility of the labor trajectories of women in Chile. Furthermore, regarding retirement, among the participants, there was a generalized perception that it would not be possible to sustain themselves economically into old age with their future pensions. As such, it is necessary for them to have a back-up plan, that is, an alternative and complementary pension strategy (R. Undurraga 2019b). However, the ability to save funds and the possibilities to opt for these various alternatives are differentiated according to their type of work trajectories, and this affects the perception that women have regarding the effects of insufficient pensions on their own old age (Gunnarsson 2018). There exist different contributory capacities for alternative pension strategies according to different types of work trajectories. The logic that operates during the work trajectory extends to retirement.

## 4. Fragile Work Trajectories under the COVID-19 Pandemic

The SARS-CoV-2 pandemic has triggered a health, economic, social, and political crisis that has shaken the foundations on which our society is based. Uncertainty and precariousness have become evident. Unemployment, social discontentment, immigration barriers, stress and anxiety, and deterioration in subjective well-being are only some of the pandemic consequences (Holmes et al. 2020; Caqueo-Urízar et al. 2020; WHO 2020). The arrival of the coronavirus has also implied important transformations in private and social life, such as the installation of telework (R. Undurraga et al. 2021), social distancing, education online, and marriages and funerals via online streaming, among other material and symbolic matters. The economic, social, and emotional effects of this pandemic have been devastating, revealing the enormous social inequalities present in Latin America, particularly with regard to gender inequalities (CEPAL 2020; ONU Mujeres/CEPAL 2020).

This crisis has not affected men in the same way as it has women. A gender analysis cannot be ignored in face of a disaster (Bradshaw 2015; Mondal 2014). Globally, when natural disasters, diseases, and crises of different kinds occur, women are the most affected, and this has also been the case for the COVID-19 pandemic; the vulnerabilities of women have clearly been exacerbated (McLaren et al. 2020; CEPAL 2020), and at a national level, Chile follows the global trend. Women have been more affected than men on various fronts; there has been a significant rise in domestic violence (Cáceres et al. 2020; López-Calva 2020; Red Chilena Contra la Violencia Hacia las Mujeres 2020), female unemployment (INE 2020b), an overload in care work and domestic tasks (Micropolíticas del Cuidado 2020), and greater stress and mental health problems (Colegio Médico de Chile 2020), among other negative consequences.

The traditional gender division of labor still operating in Chile implies that women continue to be responsible for caretaking, with important consequences in their work trajectories (R. Undurraga and Hornickel 2021). This distribution of labor has produced an overload of care work and housekeeping for women. During the pandemic, 35% of the women surveyed report having increased their working hours. They are the ones who are assuming the caretaking responsibilities both during the daytime (73% women vs. 13% men) and in the afternoon (63% women vs. 23% men) (Micropolíticas de Cuidado 2020). Likewise, the Longitudinal Study of Employment-Covid19 carried out by the UC Center for Surveys and Longitudinal Studies shows that, in times of pandemic, 38% of the men surveyed dedicated zero hours to household chores (cooking, cleaning, and washing clothes), in contrast to 14% of women; 57% of men dedicated zero hours to taking care of children under 14 years of age vs. 27.6% of women. Similarly, 71% of the men surveyed declared that they had not dedicated any hours to accompany their children in their school homework (Bravo et al. 2020).

Female employment has been particularly threatened. As a result of the pandemic, women's participation in the Chilean labor market has decreased in 10 years (INE 2020b). During 2010, the female participation rate was 46%; in 2016 it was 50.1%; in 2017 it was 52.5%, and in 2020 decreased to 41.2% (INE 2021a). This is related to the effect of the pandemic on the service sector, such as commerce, accommodation and food services, which are typically feminized activities in Chile (INE 2020b). Currently (December to February 2021 term), female labor participation is 46.3% while that of men is 69.1%; female unemployment (11%) is higher than it is for males (9.8%) (INE 2021b).

The impact of the pandemic on women's work in the present can also have long-term repercussions on their work trajectories and their old age. The differentiated work trajectories between men and women (PNUD 2010, 2014) as well as between women of different generations (Madero-Cabib et al. 2019; ComunidadMujer 2018) have implied important consequences in old age and pensions between men and women (R. Undurraga and Becker 2019). The continuing pandemic may further increase the present and future gap.

During the pandemic, citizens have demanded an early withdrawal of their pension funds as a way to alleviate the economic crisis. Two voluntary withdrawals amounting to 10% of individual pension funds have already been made and a third one is under discussion in the parliament. This has been unprecedented since the creation of the pension system administered by the AFPs. This measure reveals, among other things, the shortcomings of the Chilean Welfare State to protect citizens in the event of a catastrophe, and strongly assess the pension system and its legitimacy, underlining the current social, political, and economic crisis that the country is facing.

Below we elaborate on the possible work situations that women are facing during the current health, social, political, and economic crisis caused by the COVID-19 pandemic, according to type of trajectory.

(1) In these times of pandemic, for the women belonging to the self-realization work trajectory, the majority of them being professionals and without children or care work responsibilities, teleworking is an alternative that suits them, although isolation and loneliness can be a challenge (R. Undurraga et al. 2021). Regarding their working conditions, most of them are paid by the hour, so despite the apparent success of their careers, during the pandemic, many may be financially affected by not having a contract in case of layoffs. Their degrees and the good economic situation in which they have lived pre-pandemic do not guarantee stability or job security. In times of crisis, the fragility of this type of work trajectory is revealed.

As to their strategy of networking and having an attractive curriculum for future jobs, they could benefit from this strategy, since they are not restricted to a specific company or area, and thus, they are open to different possibilities and usually have broader networks that can be used for achieving their goals during the pandemic.

(2) Women in the assured-work trajectory could be in a different situation. To avoid COVID-19 contagions, family, education, and work have been consolidated into the same

space for 2020 and 2021 so far. The schools have closed, so children have been at home the entire academic year, requiring great involvement of the parents to take care of them, help them with online classes, and perform the daily chores without any external help; this is in addition to the hygienic and sanitary demands of the pandemic itself. Therefore, the women's total workload has substantially increased (R. Undurraga et al. 2021; Micropolíticas del Cuidado 2020). Because the vast majority of women belonging to this type of trajectory have children and caretaking responsibilities, they are possibly facing a triple burden, which could generate stress, caretaking overload, and mental exhaustion, among others. This is articulated with data showing higher levels of burnout for women (35%) than men (16%) during 2020 (Fundación Chile 2020), and higher rates of mental health issues in women than in men during the COVID-19 pandemic (Colegio Médico de Chile 2020).

Regarding their contractual situation, in the event of a layoff, they would be more protected than the rest of the population by unemployment insurance, but the high living costs are probably not covered by the amount of the insurance. In addition, another risk in the case of layoffs is that their efforts have been deposited in that workplace; therefore, their usual strategies would not serve to easily find another job since they have specialized in the guidelines and requirements of a particular company. This would be an unexpected change that affects their future. Urgent financial needs may even force them to take lower-quality jobs. In this sense, it is possible that some women are forced to change from an assured-work trajectory to an improvised one as a result of the pandemic, especially in the case of non-professional women.

(3) In the context of the pandemic, it is likely that women in the work trajectory depending on the family having paid work has decreased notably since they have to dedicate themselves to their children or other people who depend on them, without any external support (State, schools, family, private). In this case, the problem does not reside exclusively in the financial sphere; the loss of independence and personal and emotional well-being could have a high cost, since having a paid job in many cases is seen as a privilege that gives them independence and a relational world outside the home. In the context of a society that accentuates a male breadwinner/female home-caregiver model (R. Undurraga 2013), they may face the need to find a job without much experience or network due to the loss of their husbands' jobs and the impaired family economic situation. In this case, it is possible that some of the women will change to an informal situation, usually within the improvised-work trajectory, where the jobs generally do not provide positive conditions.

(4) During the coronavirus pandemic, most likely women in the improvised-work trajectory have had to continue working since the type of work they do does not allow them to work remotely. This is the case for the "lucky ones" who have kept their jobs. Many others may have had to improvise in their endeavors and take to the streets, working informally to provide for their families. Therefore, the risk of catching SARS-CoV-2 can be very serious, as they tend to live with their elderly parents in crowded homes where they could infect the at-risk group. However, faced with the multidimensionality of vulnerabilities, they sometimes must decide between working or starving and dying from the virus. The vulnerability of these women and their lack of social protection in the current context can lead to extreme poverty. This matches data from INE (2020a) showing that informality has increased during the last year and a greater number of people are living in poverty.

This type of improvised-work trajectory is a product of social and economic vulnerability. In the current context of the pandemic, the manifestation of this vulnerability has increased and, therefore, it is likely that the proportion of women in this trajectory has grown significantly.

Even though anyone is vulnerable to contracting coronavirus, not everyone would be impacted in the same way by the pandemic. An intersectional approach highlights the extent to which the intersection of social categories plays a role on exacerbating vulner-

ability. In this case, non-professional women who are head of the household, without a partner, with young children and dependents, living in a shared-rented accommodation, and belonging to the improvised-work trajectory are most likely to be in a very fragile position to support themselves and their families under the COVID-19 pandemic.

## 5. Conclusions

We have proposed an original typology of Chilean women's work trajectories, identifying four types: (1) self-realization work trajectory, (2) assured-work trajectory, (3) work trajectory depending on the family, and (4) improvised-work trajectory. Based on this typology, the article has discussed women's work trajectories and their link to Piore's theory of labor market segmentation, as well as the relation between type of work trajectory and the COVID-19 pandemic. In times of crisis, the factors that drive changes toward more insecure work trajectories become more severe, highlighting the fragility and vulnerability of female work trajectories.

The analysis of the intersection of gender, class, educational attainment, age, marital status, and number and age of children uncovers the segmentation of the Chilean labor market, delineating work paths and vulnerabilities of working life. Professional women are better off than non-professional, though educational attainment does not guarantee fair working conditions. There are still gender discriminatory laws and cultural male chauvinist work environments that put off women from the labor market. As such, women from the four types of work trajectory may experience barriers to access, maintain and advance in the labor market, however, women belonging to the improvised-work trajectory and the work trajectory depending on the family, face stronger cultural, social and economic barriers for paid work. Maternity penalties in the labor market and family penalties for working mothers are part of our participants' lives, affecting their work trajectories.

The results of the current study show a highly familiarized society and a rigid traditional gender division of labor. This means that care issues are seen as something feminine and to be resolved individually or within families. People with greater economic resources look to the private market as a way of responding to their care needs, while those with fewer resources have no choice but to exclude themselves from the labor market to care for their dependents. The help that the State offers is temporary and insufficient for the majority of Chilean women to rely on these measures and be able to work full time. Therefore, the labor trajectories of women are tremendously influenced by fluctuations related to family situations. In the current coronavirus context, this could become even more critical, due to school closings and the increase in care work and housekeeping, as the pandemic demands. These responsibilities fall mainly on the shoulders of women, hindering their participation in the labor market. Therefore, access to their own income and the consequent autonomies acquired from paid work are restricted. It is urgent to reconfigure the relationship between the State, the family, and the market in pursuit of greater gender equity.

Special attention must be paid to promoting female employment, which has decreased in relation to the 10 years prior to the year of pandemic. Strengthening the care services offered by the State is crucial; private care services are high cost, so only few families could access and afford private childcare. According to the results of this research, most women like to work; those who do not work permanently do not choose this as a personal preference primarily because they need to take care of people who depend on them (children, parents, etc.). For this reason, it is necessary to implement public policies that take into account the difficulties that most women face in participating in the labor market on a constant basis, such as offering quality child and elderly care services that are accessible for everybody. In times of crisis, like the pandemic, this becomes even more evident. The fact that the schools have been closed for so long has been brutal. The return to school should be a priority. The recent "Protect" State bonus that provides a specific amount to working mothers with children under two years of age for childcare is a move in the right direction, although the age of the child and the requirements for applying could be extended to realistically meet the need for childcare.

In Chile, most of the labor policies related to family and care issues are focused on women. For example, the benefit of Article 203 of the Labor Code on the provision of nurseries for children under two years of age is granted to working mothers only, not to working fathers. These regulations assume that the responsibility for caretaking rests with the mother. As a result of this and other gendered assumptions, women experience discrimination in the labor market (R. Undurraga 2018, 2019a; R. Undurraga and Barozet 2015). Since the costs of parenthood are not shared between genders, neither culturally nor legally, it is necessary to create more public policies focused on work-family conciliation, to be applied both to men and women. As such, it is necessary to promote joint responsibility for caretaking.

The current Chilean social security and pension systems are built on individual capitalization based on individual savings. This system grants pensions that do not allow for dignified old age; on average, the incomes of elderly are below the poverty line. This requires a serious rethinking of the pension model to integrate components of greater solidarity, such as generating a social security system, rather than individual security in which the risks fall on individuals. Although the Chilean government has promoted multiple focused measures to alleviate the economic crisis caused by the COVID-19 pandemic, these have not been enough, since the weakness lies in the type of State (liberal residual), which in times of crisis shows its fragility, but even more so, makes visible the fragility of the people in the face of a neoliberal system. Resuming the role of the State as guarantor of the citizens is key, both in terms of social protection rights and in the distribution of individual and family responsibilities. We urge the creation of public policies with greater gender equality.

This study has certain limitations. The fieldwork was carried out pre-pandemic, so the evidence used to create the typology of work trajectories is prior to the pandemic. The reflection based on this typology, in relation to the COVID-19 pandemic, contributes to a greater understanding of the work trajectories and conceives the ways in which vulnerabilities unfold in times of crisis in particular groups, specifically, women. Studying the effects of the pandemic on work and family is still an open field of study. In this sense, and for future research, it would be interesting to examine post-pandemic work trajectories of women and how the pandemic affected changes in the types of trajectories. At the same time, it would be very interesting to examine male work trajectories in connection with family and the pandemic.

The analysis of the relationship between work and family, particularly in the case of women from emerging countries, could serve to better identify the diversity and heterogeneity of their work trajectories in order to create public policies sensitive to the particularities of the population, which support the citizens in facing the health, economic, political, and social crises produced by COVID-19. This paper contributes by discussing women's work trajectories in Chile in order to grasp the effects of the health emergency generated by SARS-CoV-2 on gender inequalities at work and within families.

**Author Contributions:** Conceptualization, R.U. and J.G.; methodology, R.U. and J.G.; formal analysis, R.U. and J.G.; investigation, R.U. and J.G.; resources, R.U.; data curation, R.U. and J.G.; writing—original draft preparation, R.U. and J.G.; writing—review and editing, R.U. and J.G.; visualization, R.U.; supervision, R.U.; project administration, R.U.; funding acquisition, R.U. All authors have read and agreed to the published version of the manuscript.

**Funding:** This research was funded by Comisión Nacional de Investigación Científica y Tecnológica, Chile. CONICYT/FONDECYT, grant number 11150862.

**Institutional Review Board Statement:** The study was conducted according to the guidelines of the Declaration of Helsinki, and the Ethics Committee of the Universidad Finis Terrae approved the protocol (project number 11150862) on 22 December 2015.

**Informed Consent Statement:** Written informed consent was obtained from all subjects involved in the study.

**Data Availability Statement:** Data sharing is not applicable to this article.

**Conflicts of Interest:** The authors declare no conflict of interest.

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
