# Peer review of "The Fragility of Women’s Work Trajectories in Chile"

_socsci, doi:10.3390/socsci10050148_

Round 1
Reviewer 1 Report
Review:
The aim of the paper is to present a typology of Chilean women’s work trajectories based on a qualitative study and to reflect on the way in which the coronavirus pandemic affects these trajectories and women’s labor market participation. In the current research context, the topic of the paper has a high relevance. However, the results of the qualitative study are valid for the period before the Covid pandemic. The topic is original as far as it tries to reflect on the impact of the pandemic on Chilean women’s labor market participation, but in my opinion more detail analysis is needed based on statistical or survey data e.g., that support the reflections and the conclusions of the authors about the possible implications of the crisis. My suggestion is to reformulate the title of the paper and put stronger emphasis on the fact that the results from the study describe the pre-pandemic context and to revise the ‘Results and Discussion’ part by adding relevant recent data on the impact of the pandemic on women’s employment trajectories. In this way, authors’ reflections on the impact of the crisis on these trajectories would receive more empirical evidence.
The paper contributes as a topic to the recent literature on the multiple dimensions of Covid crisis since it highlights the perspective and the voices of Chilean women and the specifics of the socio-economic context in which they live. The paper is interesting also as an attempt to link the specifics of Chilean women’s labor market participation before the pandemic and the recent situation of profound health and economic crisis.
The paper is well written, clear and easy to read. The conclusions are consistent with the aims of the paper but again, more empirical evidence is needed that supports the reflections of the authors about the changes in the women’s employment and labor market participation in the current situation.
Author Response
Thank you for the observations.
- We reformulated the title of the paper by eliminating the association to the pandemic (p.1). We changed the abstract accordingly (p.1).
- We added empirical data on women’s employment, comparing pre-pandemic years (2010), at the time of the study (2016-2017), and the changes during the pandemic (2020 and 2021) (p.10). We added current survey data that shows women’s overload of reproductive work during the COVID-19 pandemic and the unequal distribution of labour in Chile (p.11). This empirical evidence supports our findings and reflections upon the fragility of women’s work trajectories.
- We revised and reorganized the Results and Discussion section. We differentiated the results of the study (pages 4 to 9) from a new section regarding the COVID-19 pandemic (pages 9 to 12), to make clear that the results of the study are valid for the pre-pandemic period.

Reviewer 2 Report
The article "The Fragility of Chilean Women's Work Trajectories during the COVID-19 Pandemic" tackles the important topic of vulnerability of women in the Chilean labor market. Using qualitative interviews the authors show a high level of general precariousness (as opposed to security) in the womens' work trajectories.
The article has three major flaws, but even though they are major, they can be easily fixed.
1) The role of the COVID-19 pandemic. The data were gathered between 2016 and 2017. Hence, the data should only be used to describe the vulnerability of women in the labor market during normal circumstances. All references to the pandemic must be removed, but they can be moved to an extra section that uses extrapolations for the COVID-19 pandemic, which are based on the normal situation as analyzed and described by the authors. The authors must be clear in the article that their data allow no direct analyses of the impact of the pandemic on the work trajectories of the women. Therefore I recommend to take the Covid-19 references out of the headline and the introduction. You can still announce in the abstract that you will extrapolate the impact of the pandemic on vulnerable work trajectories.
2) You have no analytical framework that explains the varying degrees of vulnerability. In explorative studies that is alright. However, there has already been some groundwork on this topic: The intersectionality approach should provide you with the perfect framework to explain why vulnerability varies between the women. Being a women is a hardship in the labor market. But the overlap of womanhood with motherhood increases vulnerability considerably. Additional layers of discrimination/vulnerability may be added and should be considered (if you have the data). For example, ethnicity/nativity, religion, age, etc.
3) Provide a theoretical ex-post explanation for the four labor market trajectories and the vulnerabilities the women will probably face. As I have read your typology of the female work trajectories, the labor market segmentation theory by Piore came to my mind. Following his theory the self-realization trajectory resembles the upper tier of the primary labor market, the assured-work trajectory the lower tier of the primary labor market. The fourth (improvised work) trajectory resembles the secondary labor market. The three segments have different meanings for the employability and job security of the women. In the first, the human capital is occupation-specific and changing the employer does not devalue the women’s' occupation-specific human capital. That means their job security might not be high, but they should have not too much trouble in finding new jobs on a comparative level. For the second trajectory the human capital is different - it is firm-specific. That means the women should enjoy a comparatively high level of job security but if they lose their job they would have a hard time finding a new one in the primary labor market. Women in the fourth trajectory are in secondary labor market. They are one of a million laborers who are easily exchangeable. Work is "simple" routine work that can be done by everybody and it needs no knowledge or extra qualification to do this kind of work. All the power in the relationship between employee and employer lies with the employer - hence you find the highest level of vulnerability.
See: Piore, M. J. (1975). Notes for a theory of labor market stratification. Labor Market Segmentation. R. C. Edwards, M. Reich and D. M. Gordon: 125-150.
The third trajectory is similar to the second but the women take an adaptive approach that allows them to reconcile their work/care preferences/roles.
As you can see - you have a lot of music in your data that, if presented correctly, would contribute greatly to the scholarly discussion on discrimination, gender roles and labor markets.
Minor issues:
A minor flaw of the article is that you give no information on the labor market institutions in Chile. You give a lot of information about the pension system, which is not so relevant for the actual work-trajectories. What about employment protection legislation, unemployment insurance, health care, social assistance, active labor market policies, etc. To what degree are the citizens of Chile independent from the labor market (decommodification)? How much economic pressure lies on them to take even the lowest of jobs available to them (you have to assume that the readers of your article are not familiar with the Chilean labor market)?
You should define vulnerability.
Based on the theoretical framework you choose (it does not have to be the one suggested by me, but you definitely need one) you should also consider to formulate hypothesis.
What happened to the retirees who did not work anymore? Do you have them excluded from your analyses? You could use the information on their completed work biography to infer the number of changes between the four types of trajectories you identified. Was there ever an upward move from type four to any other type? How did the finished work trajectories end (did they all end in type 4)?
Author Response
Thank you very much for the suggestions.
We revised and reorganized the Results and Discussion section. We differentiated the results of the study (pages 4 to 9) from a new section regarding the COVID-19 pandemic (pages 9 to 12), to make clear that the results of the study are valid for the pre-pandemic period. We amended the title, abstract, introduction, and body of the text accordingly.
We agree with intersectionality. We consider the intersection of age, gender, marital status, motherhood, and class associated with educational attainment to elaborate upon women’s work trajectories. Unfortunately, the data collected does not include information about religion, ethnicity, sexual orientation, or other social categories that could also impact women’s trajectories and their vulnerability in the labour market. We included a paragraph on the intersectional approach (p.2) to explain the intersectional analysis and bring it out on page 4.
We included Piore’s theory of labor segmentation to provide a theoretical explanation of our typology (pages 4 to 9). Thank you for the suggestion!
We included the Chilean labour market context at the very beginning of the paper to better understand women’s work trajectories (p.1). We reduced information about the pension system, included trade union restrictions, cultural factors, and a broader context of the situation of women in Chile.
We defined vulnerability on page 6.
The sample contains women from 24 to 88 years old, having and not having paid work, retired and active in the labour market (we added this information in the sample section on page 2). We included all of them in the analysis for the typology.
The article highlights that women are more vulnerable than men during their retirement because of the rules of the pension system and their trajectories. Most retired women do not rely exclusively on their pension amount; half of the non-professionals continue working, some professionals have savings and/or investments, and various participants are supported by their family (this data is discussed further in another paper). The State offers protection to women under worse circumstances –those who haven’t worked during their lifetime thus have not been able to save in the AFP, among others.
Our main hypothesis is that it is likely that many women will decrease on their type of trajectory caused by the COVID-19 pandemic, affecting their job stability and working conditions. This is articulated with data from the Institute of National Statistics, which shows that informal work and unemployment have dramatically increased, particularly for women (participation rate in the labour market and unemployment is included in the paper).
The information from biograms and interviews is not exactly a completed work biography following a chronological order. The data gathered respects the participants’ criteria to narrate their work trajectory, highlighting their milestones and relevant experiences. This tool (biogram) has an epistemological feminist standpoint; the interviewee has the power to include or exclude what they want, how they prefer to tell, and in the order they want to narrate. We found several ways to organise work trajectories: a timeline, a work path described from personal events (marriage, childbirth), descriptions following feelings and sensations about each job, turning points in life-work, and by using words, circle, lines, simple drawings, etc. The data gathered is quite rich, however, we are afraid that it is not possible to infer the exact number of changes between trajectories. Nonetheless, we can recount work trajectories based on participants’ view as follow.
Not all participants ended their trajectory in type 4 (improvised-work trajectory). Some domestic workers from the assured-work trajectory started on the improvised-work trajectory, experiencing an upward in their working conditions, job stability, and well-being. A few interviewees started and ended up on the self-realization work trajectory i.e. professionals (p.9). Though, in times of crisis like the coronavirus pandemic, the vulnerability of women’s work trajectory has emerged (pages 9 to 12).
Thank you for the suggestions!

Round 2
Reviewer 2 Report
Thanks a lot for the revised manuscript. It has improved a lot. However, a few minor details still have to be addressed.
- You write that you included the intersectionality approach as a theoretical framework for your analysis. However, it only comes up in the end of your introduction and has a very short appearance in your result section. Please use the categories gender, age, education, marital status, which you announced in the introduction, to analyze differences in the four trajectories you identified (maybe also include number of children/motherhood?). You need at least one additional paragraph in your result section to sum up the four trajectories with regard to the multiple factors that disadvantage Chilean women in the labor market. You already have one Table (table 3) that summarizes some information on the trajectories, but it should be simplified and should cover all your intersectionality categories. I propose that you use a simple descriptive table that uses relative values (per cent) for each of your intersecting social categories (education, age, marital status, motherhood) within each work trajectory. It would show that the self-realization trajectory is young, high qualified, probably not often married and mostly without children. It would probably show the complete opposite for the improvised trajectory: low education, married, with (many) children and middle aged. And so on. From that table you could derive some information about how social identities and social roles combine to create differences in labor market specific disadvantages and vulnerabilities.
- Try to improve the introduction of Piore (line 150). Maybe start from a more general perspective (labor market segmentation theory – background = explanatory approach to the discrimination of minorities in labor markets (see Kalleberg, A. L. & Sørensen, A. B. 1979. The Sociology of Labor Markets. Annual Review of Sociology, 5: 351-379.), which was developed with male workers in labor markets that were dominated by the industrial sector in mind). Point out why it helps the understanding of the vulnerability of Chilean women (discrimination, disadvantegd groups on the labor market, etc.) and then become more specific by relating to Piore’s work.
- Be more precise with your conclusions! You say that pensions are not enough to afford a living. While you are probably right, do your data corroborate this statement? Do the women in retirement age reported that they still work because they need the money? If so, you should write it out. Does it really mean the pensions are to low or do they still have to provide for a large household (would it be sufficient for them alone)? Please be specific and make clear whether your statements are driven by data or whether they are (sound) assumptions.
- The point 4. “Fragile work trajectories under the COVID-19 pandemic” needs to be a bit shortened. You start describing the general picture for women during the pandemic and then go into the specific trajectories. That is good. But you have included quite a bulk of additional information for trajectory two that is not specific to trajectory two but to all women with children. You should move the section up to the general part, shorten it and discuss its implications there. What you should concentrate on in this section is the role of the four trajectories for the labor market vulnerabilities of the women under COVID-19. Maybe you could even add a passage on how the vulnerabilities relate to the intersecting categories you analyzed.
- The conclusions need some refinement. Make sure that your conclusions link back to your analyses and not to the general problems of the Chilean welfare state. Use your intersectionality approach to systematize the sources of vulnerability and potential solutions. Start with education, because it seems to be the best way to avoid vulnerability. How is the access to higher education regulated? Is it gender- or class specific? What could be done to improve the level of education? Then you do the same with marriage and children. Here you have already some input on how to develop the relevant institutions and policies further to negate the marriage and motherhood-penalty on the labor market. What about availability, access, costs, and quality of childcare infrastructure? Then you can add the age and bring your shortened critique on the pension system.
- Please double check your English language especially in newly added passages.
Author Response
Observation 1: “You write that you included the intersectionality approach as a theoretical framework for your analysis. However, it only comes up in the end of your introduction and has a very short appearance in your result section. Please use the categories gender, age, education, marital status, which you announced in the introduction, to analyze differences in the four trajectories you identified (maybe also include number of children/motherhood?). You need at least one additional paragraph in your result section to sum up the four trajectories with regard to the multiple factors that disadvantage Chilean women in the labor market. You already have one Table (table 3) that summarizes some information on the trajectories, but it should be simplified and should cover all your intersectionality categories. I propose that you use a simple descriptive table that uses relative values (per cent) for each of your intersecting social categories (education, age, marital status, motherhood) within each work trajectory. It would show that the self-realization trajectory is young, high qualified, probably not often married and mostly without children. It would probably show the complete opposite for the improvised trajectory: low education, married, with (many) children and middle aged. And so on. From that table you could derive some information about how social identities and social roles combine to create differences in labor market specific disadvantages and vulnerabilities”.
Answer 1: We removed table 3 “Type of work trajectory according to age and educational attainment” and replaced it with a new Table 3 “Distribution (%) of age, educational attainment, marital status, and motherhood situation in each type of work trajectory”, and added a paragraph addressing the intersectionality of the categories within the different types of work trajectories (p. 8-9).
Observation 2: “Try to improve the introduction of Piore (line 150). Maybe start from a more general perspective (labor market segmentation theory – background = explanatory approach to the discrimination of minorities in labor markets (see Kalleberg, A. L. & Sørensen, A. B. 1979. The Sociology of Labor Markets. Annual Review of Sociology, 5: 351-379.), which was developed with male workers in labor markets that were dominated by the industrial sector in mind). Point out why it helps the understanding of the vulnerability of Chilean women (discrimination, disadvantaged groups on the labor market, etc.) and then become more specific by relating to Piore’s work”.
Answer 2: We started the Results and Discussion section with a broader introduction on labor market segmentation and inequalities to introduce the relevance of Piore’s theory to Chilean women’s work paths (page 4).
Observation 3: “Be more precise with your conclusions! You say that pensions are not enough to afford a living. While you are probably right, do your data corroborate this statement? Do the women in retirement age reported that they still work because they need the money? If so, you should write it out. Does it really mean the pensions are to low or do they still have to provide for a large household (would it be sufficient for them alone)? Please be specific and make clear whether your statements are driven by data or whether they are (sound) assumptions”.
Answer 3: We included a participant’s quote showing that the pension amount is not enough to afford a living so she needs to keep informally working, and added national statistics supporting this qualitative data (page 8).
Observation 4: “The point 4. “Fragile work trajectories under the COVID-19 pandemic” needs to be a bit shortened. You start describing the general picture for women during the pandemic and then go into the specific trajectories. That is good. But you have included quite a bulk of additional information for trajectory two that is not specific to trajectory two but to all women with children. You should move the section up to the general part, shorten it and discuss its implications there. What you should concentrate on in this section is the role of the four trajectories for the labor market vulnerabilities of the women under COVID-19. Maybe you could even add a passage on how the vulnerabilities relate to the intersecting categories you analyzed”.
Answer 4: We moved up data about the gendered distribution of labor during the COVID-19 pandemic (p.11). We have included that new information, as Reviewer 1 required it. We added a final paragraph in section 4 regarding intersectionality and vulnerabilities under the COVID-19 pandemic (page 13).
Observation 5: “The conclusions need some refinement. Make sure that your conclusions link back to your analyses and not to the general problems of the Chilean welfare state. Use your intersectionality approach to systematize the sources of vulnerability and potential solutions. Start with education, because it seems to be the best way to avoid vulnerability. How is the access to higher education regulated? Is it gender- or class specific? What could be done to improve the level of education? Then you do the same with marriage and children. Here you have already some input on how to develop the relevant institutions and policies further to negate the marriage and motherhood-penalty on the labor market. What about availability, access, costs, and quality of childcare infrastructure? Then you can add the age and bring your shortened critique on the pension system”.
Answer 5: We started the Conclusions slightly differently, and added a paragraph on the intersectional analysis (page 13).
At the beginning of the paper, we highlighted the relevance of cultural factors to understand the low female participation rate (p.1). We provided evidence and literature of gender discrimination both for professional and non-professional women throughout the paper (page 1, 4, 7, 10) and discussed it later on in the Conclusions (page 14). Thus, this is not only about education – professional women suffer severe discrimination in the Chilean labor market. Childcare provision is also discussed on page 14.
We kept the discussion and problems of the Chilean welfare state in relation to the current crisis as this paper is for the special issue "Family, Work and Welfare: A Gender Lens on COVID-19".
Observation 6: “Please double check your English language especially in newly added passages”.
Answer 6: A native speaker checked the English language of the paper.
